SCIENCE FORUM

# Unit of analysis issues in laboratory-based research

**Abstract** Many studies in the biomedical research literature report analyses that fail to recognise important data dependencies from multilevel or complex experimental designs. Statistical inferences resulting from such analyses are unlikely to be valid and are often potentially highly misleading. Failure to recognise this as a problem is often referred to in the statistical literature as a *unit of analysis* (UoA) issue. Here, by analysing two example datasets in a simulation study, we demonstrate the impact of UoA issues on study efficiency and estimation bias, and highlight where errors in analysis can occur. We also provide code (written in R) as a resource to help researchers undertake their own statistical analyses.
DOI: https://doi.org/10.7554/eLife.32486.001

**NICK R PARSONS[†]\*, M DAWN TEARE[†] AND ALICE J SITCH**

## Introduction

Defining the *experimental unit* is a key step in the design of any experiment. The experimental unit is the smallest object or material that can be randomly and independently assigned to a particular treatment or intervention in an experiment (*Mead et al., 2012*). The experimental unit (e.g. a tissue sample, individual animal or study participant) is the object a scientist wants to make inferences about in the wider population, based on a *sample* in the experiment. In the simplest possible experimental setting where each experimental unit provides a single outcome or observation, and *only* in this setting, the *experimental unit* is the same as both the *unit of observation* (i.e the unit described by the observed outcomes) and the *unit of analysis* (UoA) (i.e. that which is analysed). In general this will not always be the case, so care must be taken, both when planning and reporting research, to clearly define the experimental unit, and what data are being analysed and how these relate to the aims of the study.

In laboratory based research in the biomedical sciences it is almost always the case that multiple observations or measurements are made for each experimental unit. These multiple observations, which could be simple replicate measurements from a single sample or observations from multiple sub-samples taken from a

single sample, allow the variability of the measure and the stability of the experimental setting to be assessed. They improve the overall statistical power of a research study. However, multiple or repeat observations taken from the same experimental unit tend to be more similar than observations taken from different experimental units, irrespective of the treatments applied or when no treatments are applied. Therefore data within experimental units are likely to be *dependent* (*correlated*), whereas data from different experimental units are generally assumed to be *independent*, all other things being equal (i.e after removing the direct and indirect effects of the experimental interventions and setting).

The majority of widely reported statistical methods (e.g. t-tests, analyses of variance, generalized linear models, chi-squared tests) assume independence between all observations in an analysis, possibly after conditioning on other observed data variables. If the UoA is the same as the experimental unit (i.e. a single observation or summary measure is available for each unit) then the independence assumption is likely to be met. However, many studies reported in the biomedical research literature using multilevel design, often also referred to as mixed-effects, nested or hierarchical designs (*Gelman and Hill, 2007*), or more complex structured designs, fail to recognise the fact that

**\*For correspondence:** nick.parsons@warwick.ac.uk

[†]These authors contributed equally to this work

**Competing interests:** The authors declare that no competing interests exist.

independence assumptions are unlikely to be valid, and thus the reported analyses are also unlikely to be valid. Statistical inferences made from such analyses are often highly misleading.

UoA *issues*, as they are termed in the statistical literature (*Altman and Bland, 1997*), are not limited to biomedical laboratory studies, and are recognised as a major cause of concern more generally for reported analyses in bioscience and medicine (*Aarts et al., 2014*; *Altman and Bland, 1997*; *Bunce et al., 2014*; *Fleming et al., 2013*; *Lazic, 2010*; *Calhoun et al., 2008*; *Divine et al., 1992*), and also feed into widely acknowledged issues around the lack of reproducibility and repeatability of much biomedical research (*Academy of Medical Sciences, 2017*; *Bustin and Nolan, 2016*; *Ioannidis et al., 2014*; *McNutt, 2014*).

The RIPOSTE (Reducing IrreProducibility in labOratory STudiEs) framework was established to support the dialogue between scientists and statisticians in order to improve the design, conduct and analysis of laboratory studies in biomedical sciences in order to reduce irreproducibility (*Masca et al., 2015*). The aim of this manuscript, which evolved directly from a number of recommendations made by the RIPOSTE framework, is to help laboratory scientists identify potential UoA issues, to understand the problems an incorrect analysis may cause and to provide practical guidance on how to undertake a valid analysis using the open source R statistical software (*R Core Team, 2016*; *Ihaka and Gentleman, 1996*). A simple introduction to the basics of R is available from *Venables et al., 2017* and sources of information on implementation of statistical methods in the biosciences are widely available (see, for example, *Aho, 2014*).

A simulation study is undertaken in order to quantify losses in efficiency and inflation of the false positive rate that an incorrect analysis may cause (Appendix 1). The principles of experimental design are briefly discussed, with some general guidance on implemtation and good practice (Appendix 2), and two example datasets are introduced as a means to highlight a number of key issues that are widely misunderstood within the biomedical science literature. Code in the R programming language is provided both as a template for those wishing to undertake similar analyses and in order that all results here can be replicated (Appendix 3); script is available at *Parsons, 2017*. In addition, a formal mathematical presentation of the most common analysis error in this setting is also provided (Appendix 4).

## Methods and materials

### Background

A fundamental aspect of the design of all experimental studies is a clear identification of the *experimental unit*. By definition, this is the smallest object or material that can be randomly and independently assigned to a particular treatment or intervention in the experiment (*Mead et al., 2012*). The experimental unit is usually the unit of statistical analysis and should provide information on the study outcomes independent of the other experimental units. Where here the term *outcome* refers to a quantity or characteristic measured or observed for an individual unit in an experiment; most experiments will have many outcomes (e.g. expression of multiple genes, or mutiple assays) for each unit. The term *multiple outcomes* refers to such situations, but is not the same as *repeated outcomes* (or more often *repeated measures*) which refers to measuring the same *outcome* at multiple timepoints. Experimental designs are generally improved by increasing the number of (independent) experimental units, rather than increasing the number of observations within the unit beyond what is require to measure within unit variation with reasonable precision. If only a single observation of a laboratory test is obtained for each subject, data can be analysed using conventional statistical methods provided all the usual cautions and necessary assumptions are met. However, if there are for instance multiple observations of a laboratory test observed for each subject (e.g. due to multiple testing, duplicated analyses of samples or other laboratory processes) then the analysis must properly take account of this.

If all observations are treated equally in an analysis, ignoring the dependency in the data that arises from multiple observations from each sample, this leads to inflation of the false positive (type I error) rate and incorrect (often highly inflated) estimates of statistical power, resulting in invalid statistical inference (see Appendix 1). Errors due to incorrect identification of the experimental unit were identified as an issue of concern in clinical medicine more than 20 years ago, and continue to be so (*Altman and Bland, 1997*). The majority of such UoA issues involve multiple counting of measurements from individual subjects (experimental units); these issues

have particular traction in for instance orthopaedics, ophthalmics and dentistry, where they typically result from measurements on right and left hips, knees or eyes of a study participant or a series of measurements on many teeth from the same person.

The drive to improve standards of reporting and thereby design and analysis of randomized clinical trials, which resulted in the widely known CONSORT guidelines (*CONSORT GROUP (Consolidated Standards of Reporting Trials) et al., 2001*), has now expanded to cover many related areas of biomedical research activity. For instance, work by (*Kilkenny et al., 2009*) highlighted poor standards of reporting of experiments using animals, and made specific mention of the poor reporting of the number of experimental units; this work led directly to the ARRIVE guidelines (Animal Research: Reporting of In Vivo Experiments; *Kilkenny et al., 2010*) that explicitly require authors to report the study experimental unit when describing the design. The recent Academy of Medical Sciences symposium on the reproducibility and reliability of biomedical research (*Academy of Medical Sciences, 2017*) specifically highlighted poor experimental design and inappropriate analysis as key problem areas, and highlighted the need for additional resources such as the NC3Rs (National Centre for the Replacement, Reduction and Refinement of Animals in Research) free online experimental design assistant (*NC3Rs, 2017*).

### Design

The experimental unit should always be identified and taken into account when designing a research study. If a study is assessing the effect of an intervention delivered to groups rather than individuals then the design must address the issue of clustering; this is common in many health studies where a number of subjects may receive an intervention in a group setting or in animal experiments where a group of animals in a controlled environment may be regarded as a cluster. This is also the case if a study is designed to take repeated measurements from individual subjects or units, from a source sample or replicate analyses of a sample itself. Individuals in a study may also be subject to inherent clustering (e.g. family membership) which needs to be identified and accounted for.

As a prelude to discussion of analysis issues, it is important to distinguish between a number of widely reported and distinct types of data resulting from a variety of experimental designs.

The word subject is used here loosely to mean the *subject under study* in an experiment and need not necessarily be an individual person, participant or animal.

i. *Individual subjects:* In many studies the UoA will naturally be an individual subject, and be synonymous with the experimental unit. A single measurement is available for each subject, and inferences from studies comprising groups of subjects apply to the wider population to which the individual subject belongs. For example, a blood sample is collected from $n$ patients (*experimental units*) and a haemoglobin assay is undertaken for each sample. Statistical analysis compares haemoglobin levels between groups of patients, where the variability between samples is used to assess the significance of differences in means between groups of patients.

ii. *Groups of subjects:* Measurements are available for subjects. However, rather than being an individual subject, the *experimental unit* could be a group of subjects that are exposed to a treatment or intervention. In this case, inferences from analyses of variation between experimental units, apply to the groups, but not necessarily to individual subjects within the groups. For example, suppose $n \times m$ actively growing maize plants are planted together at high density in groups of size $n$ in $m$ controlled growing environments (growth rooms) of varying size and conditions (e.g. light and temperature). Chlorophyll fluorescence is used to measure stress for individual plants after two weeks of growth. Due to the expected strong competition between plants, inferences about the effects of the environmental interventions on growth are made at the *room level* only. Alternatively, in a different experiment the same plants are divided between growth rooms, kept spatially separated in *notionally* exactly equivalent conditions, after being previously given one of two different high strength foliar fertiliser treatments. Changes in plant height (from baseline) are used to assess the effect of the foliar interventions on individual plants. Although the intention was to keep growth rooms as similar as possible, inevitably *room-effects* meant that outcomes for individual plants tended to be more similar if they came from the same room, than if they came from different rooms. In this setting the plant is the *experimental unit*, but

account needs to be made for the *room-effects* in the analysis.

iii. *Multiple measurements from a single source sample:* In laboratory studies, the experimental unit is often a sample from a subject or animal, which is perhaps treated and multiple measurements taken. Statistical inferences from analyses of data from such samples should apply to the individual tissue (source) from which the sample was taken, as this is the *experimental unit*. For example, consider the haemoglobin example (i), if the assay is repeated $m$ times for each of the $n$ blood samples, then there would be $n \times m$ data values available for analysis. The analysis should take account of the fact that the replicate measurements made for each sample tell us nothing useful about the variability between samples, which are the *experimental units*.

iv. *Multiple sub-samples from a single sample:* Often a single sample from an experimental unit is sub-divided and results of assays or tests of these sub-samples yield data that provide an assessment of the variability between sub-samples. It is important to note that this is not the same as taking multiple samples from an experimental unit. The variability between experimental units is not the same as, and must be distinguished from, variability within an experimental unit and this must be reflected in the analysis of data from such studies. For example, $n$ samples of cancerous tissue (*experimental unit*) are each divided into $m$ sub-samples and lymph node assays made for each. The variability between the $m$ sub-samples, for each of the $n$ experimental units, is not necessarily the same as the variability that might have been evident if more than one tissue sample had been taken from each experimental unit. This could be due to *real* differences as the multiple samples are from different sources, or *batch-effects* due to how the samples are processed or treated before testing.

v. *Repeated measures:* One of the most important types of experimental design is the so-called repeated-measures design, in which measurements are taken on the same experimental unit at a number of time-points (e.g. on the same animal or tissue sample after treatment, at more than one occasion). These multiple measurements in time are generally assumed to be correlated and regarded as repeat measurements from an experimental unit and not separate experimental units. The likely autocorrelation between temporally related measurements from the experimental units should be reflected in the analysis of such studies. For example, height measurements for the $n \times m$ plants in (ii) could have been made at each of $t$ occasions. The $t$ height measurements are a useful means of assessing temporal changes for individual plants (*experimental unit*), such as the rate of increase (e.g. per day). However, due to the likely strong correlations, increasing the number of assessment occasions will generally add much less information to the analysis than would be obtained by increasing the number of experimental units.

Clearly many of these distinct design types can be combined to create more complex settings; e.g. plants might be housed together in batches that cause responses from the plants in the same batch to be correlated (*batch-effects*), and samples taken from the plants, divided into sub-samples, and processed at two different testing centres, possibly resulting in additional *centre-effects*. For such complex designs, it is advisable to seek expert statistical advice, however the focus in the sections discussing analysis is mainly on cases (ii), (iii) and (iv). Case (i) is handled adequately by conventional statistical analysis, and although case (v) is important, it is too large a topic to discuss in great depth here (see e.g. (*Diggle et al., 2013*) for a wide ranging discussion of longitudinal data analysis). More general design issues are discussed in Appendix 2.

### Sample size
Power analysis provides a formal statistical assessment of sample size requirements for many common experimental designs; power here is the probability (usually expressed as a percentage) that the chosen test correctly rejects the study null hypothesis, and is usually set at either 80% or 90%. Many simple analytic expressions exist for calculating sample sizes for common types of design, particular for clinical settings where methods are well developed and widely used (*Chow et al., 2008*). Power increases as the square root of the sample size $n$, so power is gained by increasing $n$ but at a diminishing rate with $n$. Also power is inversely related to the variance of the outcome $\sigma^2$, so choosing a better or more stable outcome or assay or test procedure will increase power.

For the most simple design with a normally distributed outcome, comparing two groups of $n$

subjects (e.g. as in Design case (i)), the sample size is given by $n = 2\sigma^2 \times \{(z_{\alpha/2} + z_\beta)^2/d^2\}$, where $d$ is the difference we wish to detect, $z_\beta$ represents the the upper $100 \times \beta$ standard normal centile, and $1 - \beta$ is the power and $\alpha$ the significance level; for the standard significance of 5% and power of 90%, $(z_{\alpha/2} + z_\beta)^2 = (1.96 + 1.28)^2 \approx 10.5$.

Where there are clusters of subjects (e.g. as in Design case (ii)), then the correlation between observations within clusters will have an impact on the sample size (*Hemming et al., 2011*). The conventional sample size expression needs to be inflated by a variance inflation factor (VIF), also called a *design effect*, given by $VIF = 1 + (m - 1) \times ICC$, where there are $m$ observations in each cluster (e.g. a batch) and ICC is the intraclass (within cluster) correlation coefficient that quantifies the strength of association between subjects within a cluster. The ICC can either be estimated from pilot data or from previous studies in the same area (see examples), or otherwise a value must be assumed. For small cluster sizes ($m<5$) and intraclass correlations ($ICC<0.01$), the sample size needs only to be inflated by typically less than 10% (see *Table 1*). However for larger values of both $m$ and ICC, sample sizes may need to be doubled, trebled or more to achieve the required power.

For more complex settings, often the only realistic option for sample size estimation is simulation. Raw data values are created from an assumed distribution (e.g. multivariate normal distribution with known means and covariances) using a random number generator, and the planned analysis performed on these data. This process can be repeated many (usually thousands of) times and the design characteristics (e.g. power and type I error rate) calculated for various sample sizes. This has typically been a task that requires expert statistical input, but increasingly code is available in R to make this much easier (*Green and MacLeod, 2016*;

*Johnson et al., 2015*). Many application area dependent rules of thumb exist when selecting a sample size, the most general being the *resource equation* approach of (*Mead et al., 2012*), which suggests that approximately 15 degrees of freedom are required to estimate the error variance at each level of an analysis.

### Analysis

Incorrect analysis of data that have known or expected dependencies leads to inflation of the false positive rate (type I error rate) and invalid estimates of statistical power, leading to incorrect statistical inference; a simulation study (Appendix 1) shows how various design characteristics can affect the properties of a hypothetical study. Focussing on linear statistical modelling (*McCullagh and Nelder, 1998*), which is by far the most widely used methodology for analysis when reporting research in the biomedical sciences, there are generally two distinct approaches to analysis when there are known UoA issues (*Altman and Bland, 1997*).

#### Subject-based analysis

The simplest approach to analysis is to use a single observation for each subject. This could be achieved by selecting a single representative observation or more usually by calculating a summary measure for each subject. The summary measure is often the mean value, but could be for instance the area under a response curve or the gradient (rate) measure from a linear model. Given that this results in a single observation for each subject, analysis can proceed using the summary measure data in the conventional way using a generalized linear model (GLM; (*McCullagh and Nelder, 1998*)) assuming independence between all observations.

A GLM relates a (link function) transformed *response* variable to a linear combination of *explanatory* variables via a number of model parameters that are estimated from the observed data. The explanatory variables are so-

**Table 1.** Variance inflation factors for cluster sizes ($m$) 2, 5, 10 and 20, and intraclass correlation coefficients (ICC) 0.01, 0.05, 0.1 and 0.5.

| $m$ | ICC | | | |
|---|---|---|---|---|
| | **0.01** | **0.05** | **0.1** | **0.5** |
| **2** | 1.01 | 1.05 | 1.10 | 1.50 |
| **5** | 1.04 | 1.20 | 1.40 | 3.00 |
| **10** | 1.09 | 1.45 | 1.90 | 5.50 |
| **20** | 1.19 | 1.95 | 2.90 | 10.50 |

DOI: https://doi.org/10.7554/eLife.32486.002

called fixed-effects that represent the (systematic) observed data that are used to model the response variable. The lack of model fit is called the *residual* or *error*, and represents unstructured deviations from the model predictions that are beyond control. The subject-based approach is valid but has the disadvantage that not all of the available data are used in the definitive analysis, resulting in some lack of efficiency. Care must be taken when choosing a single measure for each subject, ensuring the selection does not introduce bias and if a summary measure is generated, this value must be meaningful and if appropriate the analysis should be weighted to account for the precision in estimation of the summary measure.

### Mixed-effect analysis

A better approach than the subject-based analysis, is a mixed-effect analysis (*Galwey, 2014*; *Pinheiro and Bates, 2000*). A (generalized) linear mixed effects model (GLME) is an extension of the conventional GLM, where structure is added to the *error* term, leaving the systematic fixed terms unchanged, by adding so-called random-effect terms that partition the *error* term into a set of structured (often nested) terms. In the simplest possible setting (*Bouwmeester et al., 2013*), the *error* term is replaced by a *subject-error* term to model the variation between subjects and a *within-subject error* term to model the within subject variation. This partition of the error into multiple strata allows, for instance, the correct variability (*subject-error* term) to be used to compare groups of subjects. Random-effects are often thought of as terms that are not of direct inferential interest (in contrast to the fixed-effects) but are such that they need to be properly accounted for in the model; e.g. a random selection of subjects or centres in a clinical trial, shelves in an incubator that form a temperature gradient or repeat assays from a tissue sample.

The algorithms used to estimate the model terms for a GLME and details of how to model complex *error* structures will not be discussed further, but more details can be found in for instance *Pinheiro and Bates, 2000*. Mixed-effects models can be fitted in most statistical software packages, but the focus here is on the R open source statistical software (*R Core Team, 2016*). Detailed examples of implementation and code are provided in Appendix 3 and a script is available at *Parsons, 2017* to reproduce all the analysis shown here using the R packages

nlme (*Pinheiro et al., 2016*) and lme4 (*Bates et al., 2015*).

## Results

In order to better appreciate the importance of UoA issues, to understand how these issues arise and to show statistically how analyses should be implemented, two example datasets from real experiments are described and analysed in some detail. The aims of the experiments are clearly not of direct importance, but the logic, process and conduct of the analyses are intended to be sufficiently general in nature so as to elucidate many key problematic issues.

### Example 1: Adjuvant radiotherapy and lymph node size in colorectal cancer

Six subjects diagnosed with colorectal cancer, after confirmatory magnetic resonance imaging, underwent neoadjuvant therapy comprising of a short course of radiotherapy (RT) over one week prior to resection surgery. These subjects were compared with six additional cancer subjects, of similar age and disease severity, who did not receive the adjuvant therapy. The aim of the study was to assess whether the therapy reduced lymph node size in the resection specimen (i.e. the sample removed during surgery). The resection specimen for each subject was divided into two sub-samples after collection, and each was fixed in formalin for 48-72 hr. These sub-samples were processed and analysed at two occasions, by different members of the laboratory team. The samples were sliced at 5mm intervals and images captured and analysed in an automated process that identified lymph node material which was measured by a specialist pathologist to give a measure of individual lymph node size (i.e. diameter), based on assumed sphericity. Three slices per sub-sample were collected for each subject. *Table 2* shows the measured lymph node sizes in mm for each sample.

### Naive analysis

The simplest analysis and the one that may appear to be correct if no information on the design or data structure shown in *Table 2* were known, would be a t-test that compares the mean lymph node size between the RT groups. This shows that there is reasonable evidence to support a statistically significant difference in mean lymph node size between those subjects who received RT (Short RT) and those who did not (None); mean in group None = 2.403 mm

and in group RT Short = 2.120 mm, difference in means = 0.283 mm (95% CI; 0.057 to 0.508), with a t-statistic = 2.501 on 70 degrees of freedom, and a p-value = 0.015. The conclusion from this analysis is that lymph node sizes were statistically significantly smaller in the group that had received adjuvant RT. Why should the veracity of this result be questioned?

The assumptions made when undertaking any statistical analysis must be considered carefully. The t-statistic is calculated as the absolute value of the difference between the group means, divided by the pooled standard error of the difference (sed) between the group means. This latter quantity is given by $sed = s \times \sqrt{(1/n_1 + 1/n_2)}$, where $n_1$ and $n_2$ are the sample sizes in the two groups and $s^2$ is the pooled variance given by $s^2 = ((n_1 - 1)s_1^2 + (n_2 - 1)s_2^2)/(n_1 + n_2 - 2)$; where $s_1^2$ and $s_2^2$ are the variances within each group. The important thing to realize here is that the variances within each of the RT groups are calculated by simply taking the totality of data for all six subjects in each group, across all sample types and slices. One of the key assumptions of the t-test is that of *independence*. Specifically, this requires the lymph node sizes to be all independent of each other; i.e. the observed size for one particular node is not systematically related to the other lymph node size data used for the statistical test. What is meant by *related to* in this context?

It seems highly likely that the lymph node sizes for repeat slices for any particular sample for a subject are more similar than size measurements from other subjects. Similarly, it might be expected that lymph node sizes for the two samples for each subject are more similar than lymph nodes size measurements from other subjects. If the possibility that this is important is ignored, and a t-test is undertaken, then the variability measured between samples and between slices within samples is being used to assess differences between subjects. If the assumption of independence is not valid, then by ignoring this, claims for statistical significance may be being made that are not supported by the data (See Appendix 4 for a mathematical description of the *naive analysis*).

## Subject-based analysis

Given that the lymph node size measurements within samples and subjects are likely to be more similar to each other than to data from other subjects, how should the analysis be conducted? Visual inspection of the data can often reveal patterns that are not apparent from tabular summaries; *Figure 1* shows a strip plot of the data from *Table 2*.

It is clear, from a visual inspection alone of *Figure 1*, that data from repeat slices within samples are more similar (clustered together) than data from the repeat samples within each subject. And also that data from the multiple samples and slices for each subject are generally clustered together; data from a single subject are usually very different from other subjects,

**Table 2.** Lymph node sizes (mm), by sample slice and subject, by radiotherapy (RT) group, subjects 1 to 6 no RT and subjects 7 to 12 short RT; highlighted cells are those removed to unbalance the design.

| None | | | | | | Short RT | | | | | |
|---|---|---|---|---|---|---|---|---|---|---|---|
| Subject | Sample | Slice | | | | Subject | Sample | Slice | | | |
| | | 1 | 2 | 3 | | | | 1 | 2 | 3 | |
| 1 | 1 | 1.71 | 1.98 | 1.88 | | 7 | 1 | 2.37 | 2.36 | 2.20 | |
| | 2 | 1.72 | 1.98 | 1.85 | | | 2 | 2.36 | 2.62 | 2.60 | |
| 2 | 1 | 2.51 | 2.55 | 2.65 | | 8 | 1 | 1.33 | 1.35 | 1.15 | |
| | 2 | 2.98 | 3.20 | 2.80 | | | 2 | 1.90 | 1.87 | 1.85 | |
| 3 | 1 | 1.69 | 1.72 | 1.80 | | 9 | 1 | 1.70 | 1.78 | 1.78 | |
| | 2 | 1.82 | 1.97 | 1.73 | | | 2 | 2.07 | 1.76 | 1.85 | |
| 4 | 1 | 1.72 | 1.78 | 2.04 | | 10 | 1 | 2.23 | 2.14 | 2.21 | |
| | 2 | 2.50 | 2.65 | 2.77 | | | 2 | 2.50 | 2.33 | 2.16 | |
| 5 | 1 | 3.32 | 3.27 | 3.07 | | 11 | 1 | 2.10 | 1.89 | 1.75 | |
| | 2 | 3.11 | 3.03 | 3.11 | | | 2 | 2.11 | 2.16 | 2.12 | |
| 6 | 1 | 2.33 | 2.48 | 2.53 | | 12 | 1 | 2.58 | 2.54 | 2.59 | |
| | 2 | 2.86 | 2.87 | 2.52 | | | 2 | 2.77 | 2.65 | 2.60 | |

DOI: https://doi.org/10.7554/eLife.32486.003

irrespective of the RT grouping. One, albeit crude, solution to such issues is to calculate a summary measure for each of the experimental units at the level at which the analysis is made, and use these measures for further analysis. The motivation for doing this is that it is usually reasonable to assume that experimental units (subjects) are independent of one another, so if a t-test is undertaken on summary measures from each of the twelve subjects it is also reasonable to assume that the necessary assumption of independence is true.

Using the mean lymph node size for each subject as the summary measure (subjects 1 to 12; 1.85, 2.78, 1.79, 2.24, 3.15, 2.60, 2.42, 1.57, 1.82, 2.26, 2.02, and 2.62 mm), a t-test shows

that there is no evidence to support a statistically significant difference in mean lymph node size between those subjects who received RT (Short RT) and those who did not (None); mean in group None = 2.403 mm and in group RT Short = 2.120 mm, difference in means = 0.283 mm (95% CI; -0.321 to 0.886), with a t-statistic = 1.043 on 10 degrees of freedom, and a p-value = 0.322. Note that the group means are the same but now the t-statistic is based on 10 degrees of freedom, rather than the 70 of the naive analysis, and the confidence interval is considerably wider than that estimated for the naive analysis. The conclusion from this analysis is that there is no evidence to support a difference in lymph node size between groups. Why is the

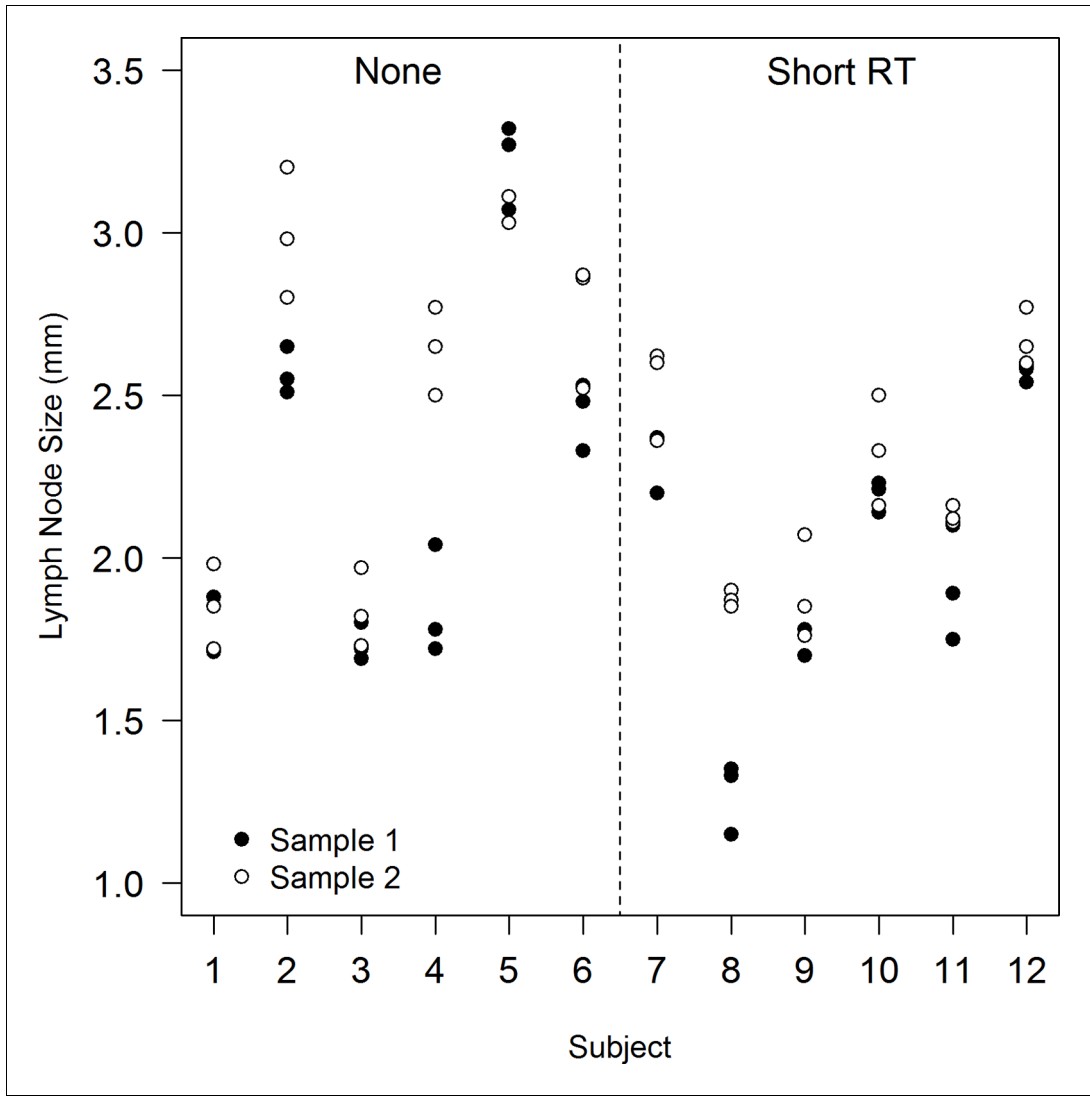

**Figure 1.** A strip plot showing observed lymph node size data by subject (1-12) and sample, after none and a short course of radiotherapy (Short RT).
DOI: https://doi.org/10.7554/eLife.32486.004

result of this t-test so different from the previous naive analysis?

In the naive analysis the variability between measurements within the main experimental units (subjects) and the variability between experimental units was used to assess the difference between experimental units. In the analysis in this section the variability between experimental units alone has been used to assess the effect of the intervention applied to the experimental units. The multiple measurements within each experimental unit improve the precision of the estimate of the unit mean, but provide no information on the variability between units, that is important in assessing interventions that are applied to the experimental units. This analysis is clearly an improvement on the naive analysis, but it uses only summary measures for each experimental unit, rather than the full data, it tells us nothing about the relative importance of the variability between subjects, between samples and between slices and it does not allow us to assess the importance of these design factors to the conclusions of the analysis.

Linear mixed-effects analysis

To correctly explain and model the lymph node data a linear mixed-effects model must be used. The experimental design used in the lymph node study provides the information needed to construct the random-effects for the mixed-effects model. Here there are multiple levels within the design that are naturally nested within each other; samples are nested within subjects, and slices are nested within samples. Fitting such a mixed-effects model gives the following estimate for the intervention effect (RT treatment groups); difference in means = 0.283 mm (95% CI; -0.321 to 0.886), with a p-value = 0.322 (t-statistic = 1.043 on 10 degrees of freedom). For a balanced design, intervention effect estimates for the mixed-effects model are equivalent to those from the subject-based analysis. A balanced design is one where there are equal numbers of observations for all possible combinations of design factor levels; in this example there are the same number of slices within samples and samples within subjects.

The mixed effects model allows the variability within the data to be examined explicitly. Output from model fitting also provides estimates of the standard deviations of the random effects for each level of the design; these are for subjects, $\sigma_P$ = 0.436 (95% CI; 0.262 to 0.727), samples $\sigma_S$ = 0.236 (95% CI; 0.151 to 0.362) and residuals (slices) $\sigma_\epsilon$ = 0.122 (95% CI; 0.100 to

0.149). Squaring to get variances, indicates that the variability, in lymph node size, between subjects was three and half times more than the variability between samples, and nearly thirteen times as much as the variability between repeat slices within samples. The intraclass correlation coefficient measures the strength of association between units within the same group; for subjects $ICC_P$ = 0.733, where $ICC_P = \sigma_P^2/(\sigma_P^2 + \sigma_S^2 + \sigma_\epsilon^2)$. This large value, which represents the correlation between two randomly selected observations on the same subject, shows why the independence assumption required for the naive analysis is wrong (i.e. independence implies that ICC = 0). This demonstrates clearly why pooling variability without careful thought about the sampling strategy and design of an experiment is unwise, and likely to lead to erroneous conclusions.

Various competing models for random effects can be compared using likelihood ratio tests (LRT). For instance in this example suppose that the two samples collected for the same subject had been arbitrarily labelled as *sample 1* and *sample 2*, and in practice there was no real difference in the methods used to process or capture images of nodes from the two samples. In such a setting, a more appropriate random effects model may be to have a subject effect only and ignore the effects of samples within subjects. Constructing such a model and comparing to the more complex model gives a LRT = 39.92 and p-value < 0.001, providing strong support in favour of the full multilevel model. Diagnostic analyses can be undertaken after fitting mixed-effects model, in an analogous manner to linear models (*Fox et al., 2011*).

*Figure 2* shows boxplots of residuals for each subject and a quantile-quantile plot to assess Normality of the residuals. Inspection of the residual plots for the lymph node size data, show that assumptions of approximate Normality are reasonable; e.g. the quantile-quantile plot of the residuals from the model fit fall (approximately) along a straight line when plotted against theoretical residuals from a Normal distribution. If residuals fail to be so well behaved and deviate in a number of well understood ways, or if for instance variances are non-equal or vary with the outcome (heterogeneity), then transforming the data prior to linear mixed-effects analysis can improve the situation (*Mangiafico, 2017*). However, in general, if the Normality assumption is not sustainable, data are better analysed using generalized linear mixed effects models (*Pinheiro and Bates, 2000*;

*Galwey, 2014*), that better account for the distributional properties of the data.

Unbalanced data analysis

Intervention effect estimates for the mixed-effects and subject-based analyses presented here are equivalent, due to the balanced nature of the design. Every subject has complete data for all samples and slices. By calculating means for each subject averaging occurs across the same mix of samples and slices, so irrespective of the effects on the analysis of these factors, the means will be directly comparable and estimated with equivalent precision. Whilst balance

is a desirable property of any experimental design, it is often unrealistic and impractical to obtain data structured in this way; for instance in this example, samples may be contaminated or damaged during processing or insufficient material may be available for all three slices.

Repeating the above mixed-effects analysis after randomly removing 50% of the data (see *Table 2*), gives an estimated difference in lymph node size between groups = 0.263 mm (95% CI; -0.397 to 0.922), with a p-value = 0.391, and estimates of the standard deviations of the random effects for each level of the design, $\sigma_P = 0.421$ (95% CI; 0.224 to 0.794), $\sigma_S = 0.279$ (95%

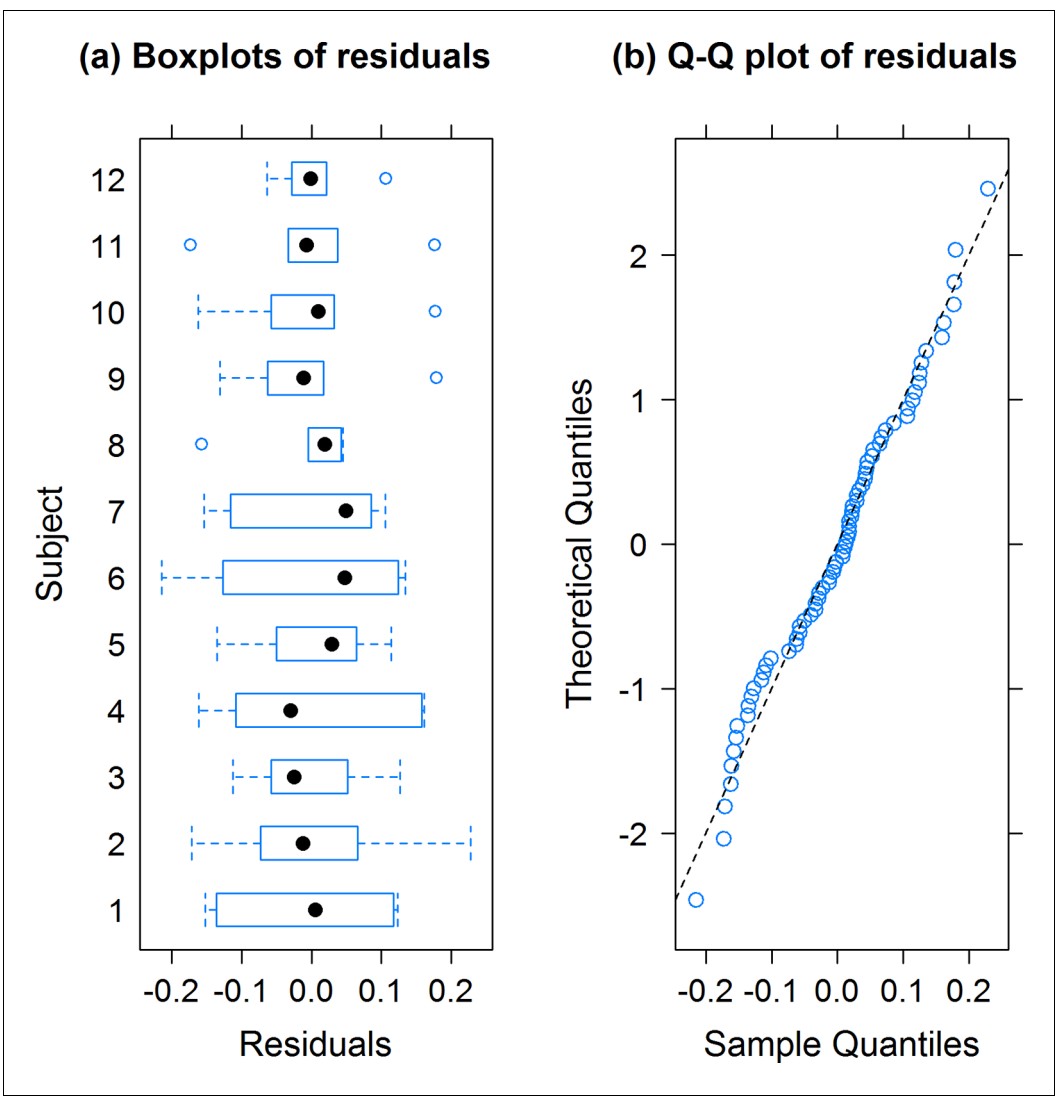

**Figure 2.** Boxplots of residuals (observed values - fitted values) for each subject; symbols (•) are medians, boxes are interquartile ranges (IQR), whiskers extend to 1.5×IQR and symbols (○) outside these are suspected outliers (a). Quantile-quantile (Q–Q) plot of the model residuals (○) on the horizontal axis against theoretical residuals from a Normal distribution on the vertical axis (b).

DOI: https://doi.org/10.7554/eLife.32486.005

CI; 0.160 to 0.489) and $\sigma_\epsilon = 0.124$ (95% CI; 0.088 to 0.174). These are, perhaps surprisingly given that only half the data from the previous analysis are being used, very similar to estimates from the complete data. However, in the unbalanced setting the subject-based analysis is no longer valid, as it ignores the variation in sample sizes between subjects; the estimated difference in lymph node size between groups is 0.199 mm (95% CI; -0.474 to 0.872) for the subject-based analysis.

### Example 2: Lymph node counts after random sampling

The most extreme example of non-normal data is for binary responses, which generally results from yes/no or present/absence type outcomes. Extending the lymph node example, in a parallel study, rather than measure the sizes of selected nodes or conduct a time-consuming count of all nodes, a random sampling strategy was used to select regions of interest (RoI) in which fives nodes were randomly selected and compared to a 2mm reference standard (≥2mm; yes or no). This could be done rapidly by a non-specialist. Five samples were processed for each of twelve subjects, in an equivalent design to the lymph node size study; data are shown in *Table 3*.

#### Non-normal data analysis

For some subjects there was insufficient tissue for five samples, resulting in an unbalanced design. The odds of an event (i.e. observing or not observing a lymph node with diameter ≥2mm), is the ratio of the probabilities of the two possible states of the binary event, and the odds ratio is the ratio of the odds in the two groups of subjects (e.g. those receiving either None or Short RT). A naive analysis of these data suggest an estimate of the odds ratio of (43/82)/(79/46) = 0.31, for RT Short versus None groups; 43 lymph nodes with maximum diameters ≥2mm from 125 in the RT Short group versus 79 from 125 in the None group. Being in the RT Short group results in a lower odds of lymph nodes with diameters ≥2mm. This is the result one would obtain by conventional logistic regression analysis; odds-ratio 0.31 (95% CI; 0.18 to 0.51; p-value < 0.001) providing very strong evidence that lymph node diameters were lower in the RT Short group.

In logistic regression analysis the estimated regression coefficients are interpreted as log odds-ratios, which can be transformed to odds ratios using the exponential function (*Hosmer et al., 2013*). However, one should be instinctively cautious about this result, as it is clear from *Table 3* that variation within subjects is much less than between subjects; i.e. some subjects have low counts across all samples and others have high counts across all samples. The above analysis ignores this fact and pools variation between samples and between subjects to test for differences between two groups of subjects. This is clearly not a good idea.

Fitting a GLME model with a subject random effect, gives an estimated odds-ratio for the Short RT group of 0.26 (95% CI; 0.09 to 0.78; p-value = 0.016). The predicted probability of detecting a lymph node with a diameter ≥2mm was 0.65 for the None RT group and 0.33 for the Short RT. The overall conclusions of the study have not changed, however the level of significance associated with the result is massively overstated in the simple logistic regression, due to the much smaller estimate of the standard error of the log odds-ratio (0.264 for logistic regression versus 0.564 for the mixed-effects logistic regression). By failing to properly account for the difference in variability between

**Table 3.** Number of five selected lymph nodes with maximum diameters ≥2mm, for up to five tissue samples per subject (1-12), after either none or a short course of radiotherapy (Short RT).

| None | | | | | | Short RT | | | | | |
|---|---|---|---|---|---|---|---|---|---|---|---|
| Subject | Sample | | | | | Subject | Sample | | | | |
| | 1 | 2 | 3 | 4 | 5 | | 1 | 2 | 3 | 4 | 5 |
| 1 | 4 | 4 | - | - | - | 7 | 1 | 0 | 0 | 0 | 0 |
| 2 | 3 | 4 | 5 | 2 | - | 8 | 1 | 2 | - | - | - |
| 3 | 2 | 3 | 3 | 2 | - | 9 | 1 | 0 | 1 | 0 | 2 |
| 4 | 2 | 4 | 1 | 2 | 1 | 10 | 2 | 1 | 4 | 0 | 2 |
| 5 | 3 | 4 | 4 | 3 | 5 | 11 | 4 | 2 | 4 | 3 | 3 |
| 6 | 2 | 5 | 5 | 3 | 3 | 12 | 3 | 4 | 3 | - | - |

DOI: https://doi.org/10.7554/eLife.32486.006

measurements made on the same subject relative to the variability in measurements between subjects results in overoptimistic conclusions.

## Discussion

The examples, simulations and code provided highlight the importance of correctly identifying the UoA in a study, and show the impact on the study inferences of selecting an inappropriate analysis. The simulation study (Appendix 1) shows that the false positive rate can be extremely high and efficiency very low if analyses are undertaken that do not respect well known statistical principles. The examples reported are typical of studies in the biomedical sciences and together with the code provide a resource for scientists who may wish to undertake such analyses (Appendix 3). Although clearly discussion with a statistician, at the earliest possible stage in a study, should always be strongly encouraged, in practice this may not be possible if statisticians are not an integral part of the research team. The RIPOSTE framework (*Masca et al., 2015*) called for the prospective registration (*Altman, 2014*) and publication of study protocols for laboratory studies, which we believe if implemented would go a long way towards addressing many of the issues discussed here by causing increased scrutiny at all stages of an experimental study.

The examples, design and analysis methods presented here have deliberately used terminology such as *experimental unit*, *subject* and *sample* to make the arguments more comprehensible, particularly for non-statisticians, who often find these topics conceptually much easier to understand using such language. This may have contributed to the widespread belief amongst many laboratory scientists that these issues are important only in human experimentation. Where, for instance, the subject is a participant in a clinical trial and the idea that subjects provide data that are independent of one another, but correlated within a subject seems perfectly natural. However, although such language is used here, it is important to emphasise that the issues discussed apply to *all* experimental studies and are arguably likely to be more not less important for laboratory studies than for human studies. The lack of appreciation of the importance of UoA issues in laboratory science may be due to the misconception that the within subject associations observed for human subjects arise mainly from the subjective nature of the measures used in clinical trials on human

subjects; e.g. patient-reported outcomes. Contrasting these with the more objective (hard) measures that dominate in much biomedical laboratory based science leads many to assume that that these issues are not important when analysing data and reporting studies in their own research area.

Mixed-effects models are now routinely used in the medical and social sciences (where they are often known as multilevel models), to for instance allow for the clustering in patient data from a recruiting centre in a clinical trial, or to model the association in outcomes within schools and classrooms from students (*Brown and Prescott, 2015*; *Snijders and Bosker, 2012*). Mixed-effects models originated from the work of pioneering statistician/geneticist R. A. Fisher (*Fisher, 1919*), whose classic texts on experimental design have led to their extensive and very early use in agricultural field experimentation (*Mead et al., 2012*). However, the use of mixed-effects models in the biological sciences has not spread from the field to the laboratory.

Mixed-effects models are not used as widely in biomedical laboratory studies as in many other scientific disciplines, which is a concern, as given the nature of the experimental work reported one would expect these models to be equally widely used and reported as they are elsewhere. This is most likley simply a matter of lack of knowledge and convention; if colleagues or peers do not routinely use these methods then why should I? By highlighting the issue and providing some guidance the hope is that this article may address the first of these issues. Journals and other interest groups (e.g. funding bodies and learned societies) have a part to play also, particularly in ensuring that work is reviewed by experienced and properly qualified statisticians at all stages from application to publication (*Masca et al., 2015*).

### Acknowledgements

This work is supported by the NIHR Statistics Group (https://statistics-group.nihr.ac.uk/). NIHR had no role in the design and conduct of the study, or the decision to submit the work for publication.

**Nick R Parsons** Warwick Medical School, University of Warwick, Coventry, United Kingdom
nick.parsons@warwick.ac.uk
http://orcid.org/0000-0001-9975-888X

**M Dawn Teare** Sheffield School of Health and Related Research, University of Sheffield, Sheffield, United Kingdom

**Alice J Sitch** Institute of Applied Health Research, College of Medical and Dental Sciences, University of Birmingham, Edgbaston, Birmingham, United Kingdom

https://orcid.org/0000-0001-7727-4497

*Author contributions:* Nick R Parsons, M Dawn Teare, Alice J Sitch, Conceptualization, Writing—original draft, Writing—review and editing, Analysis and interpretation of data

*Competing interests:* The authors declare that no competing interests exist.

## Funding

The authors declare that there was no funding for this work.

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

# Appendix 1

DOI: https://doi.org/10.7554/eLife.32486.007

## Simulation study: Demonstrating UoA issues

Consider a small hypothetical study that aims to compare outcomes from subjects randomly allocated to two contrasting treatment options, A and B. Samples were collected from subjects and detailed laboratory work undertaken to provide 24 outcome measurements for each of the two groups. For treatment group A, a measurement was obtained from 24 individual subjects; measurements for group A are known to be uncorrelated, i.e. independent of one another. However, for treatment group B no such information was available. How would the sampling strategy for group B impact on the analysis undertaken and how could it affect the interpretation of the results of the analysis?

Consider the following possibilities; (i) the sampling strategy used for treatment group B was the same as treatment group A (i.e. 24 independent samples), (ii) in group B 2 measurements were available from each of 12 subjects, (iii) 4 measurements were available from each of 6 subjects, (iv) 6 measurements were available from each of 4 subjects, (v) 8 measurements were available from each of 3 subjects and (vi) 12 measurements were available from each of 2 subjects.

Experience from previous studies suggests that the measurements made on the same individual subjects are likely to be positively correlated; i.e. if one measurement is large then the others will also be large, or conversely if one measurement is small others will also be small.

Assume for the ease of illustration that the measurements were Normally distributed, and of equal variance in each treatment group, and analyses were made using an independent samples t-test, at the 5% level. One key characteristic that is important here is the false positive rate (type I error rate); i.e. the probability of incorrectly rejecting the null hypothesis. Here the null hypothesis is that the sample mean from treatment groups A and B are the same. Figure 1(a) shows the type I error rates, based on 100000 simulations, for comparison of groups A and B, where the null hypothesis is known to be true, for scenarios (i) - (vi) for within subject correlations $\rho = 0$, $\rho = 0.2$, $\rho = 0.5$ and $\rho = 0.8$. If data within subjects are uncorrelated ($\rho = 0$), then the type I error rate is maintained at the required 5% level over all scenarios (i) to (vi), and clearly in scenario (i), where there are 24 single samples in group B, it makes no sense to consider within subject correlations as there is only a single measurement for each subject, the type I error rate is controlled at the 5% level. Otherwise, as the number of subjects gets smaller (greater clustering) and the correlation within subjects gets larger, the type I error rate increases rapidly. In the extreme scenario where there are data from 2 subjects only, with a high correlation ($\rho = 0.8$) the null hypothesis is incorrectly rejected approximately 45% of the time.

If grouped data are naively analysed, ignoring likely strong associations between measurements within the same group, it is very likely that incorrect inferences are made about differences between treatment groups.

If the true grouping structure in B were known, then how might this be properly accounted for in the analysis? One simple option to improve on the naive analysis, of assumed independence, is to randomly select a single value from each subject; this will control the type I error rate at the required level across all scenarios and correlations (*Figure 1b*), but will provide rather inefficient estimates of the treatment difference between groups (*Figure 1c*).

An alternative simple strategy is to calculate the within-subject means, this provides an unduly conservative (type I error rate $\leq$5%) test (*Figure 1b*), as the true variability in the data is typically underestimated by using the subject means. However, the analysis based on subject means rather than randomly selected values provides more efficient estimates of the treatment difference between groups (Figure 1(c)), with the efficiency depending on the within subject correlation; as the correlation within subjects increases then the value of calculating a mean, in preference to selecting a single value for each subject, diminishes markedly.

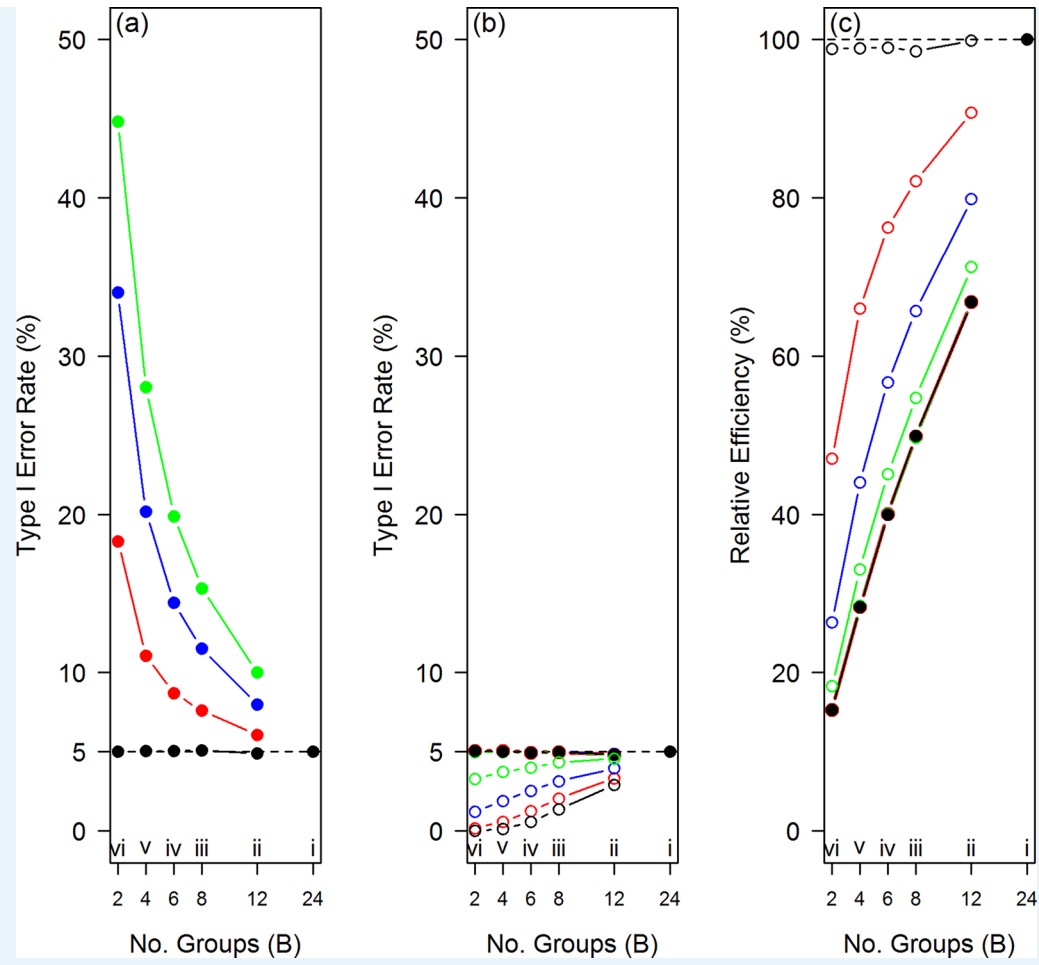

**Appendix 1—figure 1.** Naive use of a conventional t-test on correlated (grouped by subject) data, $\rho = 0$ (black circle ), $\rho = 0.2$ (red circle) $\rho = 0.5$ (blue circle) and $\rho = 0.8$ (green circle), inflates the type I error rate (set at 5%). (**a**). The type I error rate can be controlled to the required level by randomly selecting a single measurement for each subject, $\rho = 0$ (black circle), $\rho = 0.2$ (red circle), $\rho = 0.5$ (blue circle) and $\rho = 0.8$ (green circle), or made conservative ($\leq 5\%$) by taking the mean of the measurements for each subject, $\rho = 0$ (black open circle), $\rho = 0.2$ (red open circle), $\rho = 0.5$ (blue open circle) and $\rho = 0.8$ (green open circle) (**b**). The relative efficiency of treatment effect estimates declines as the number of clusters become smaller and is always higher for the mean than the randomly selected single measurement strategy (**c**). The scenarios (i) – (vi) are as described in the text.
DOI: https://doi.org/10.7554/eLife.32486.008

## Appendix 2

DOI: https://doi.org/10.7554/eLife.32486.009

### Some fundamental principles of experimental design

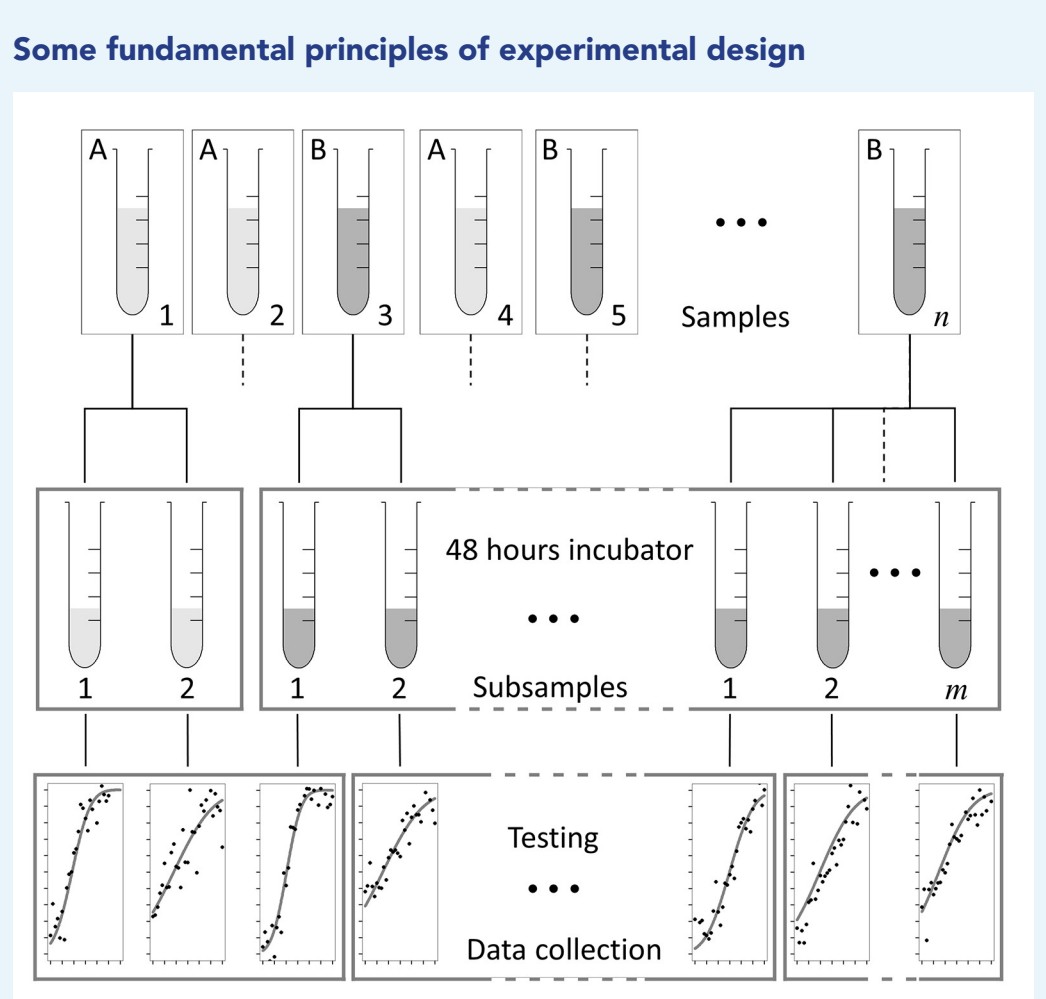

**Appendix 2—figure 1.** Design options for a putative laboratory study testing $n$ samples of experimental material.

DOI: https://doi.org/10.7554/eLife.32486.010

Consider a putative study (**Figure 1**), where $n$ samples (*experimental units*) of material are available for experimentation. Interventions (A and B) are assigned to the experimental units and sub–samples collected for processing and incubation prior to final testing 48 hours later. The scientist undertaking the study has control over the sampling strategy and the design; e.g. how to allocate samples to A and B, whether to divide samples and how to split material between incubators and the testing procedures used for data collection. What are the key issues that they need to consider before proceeding to do the study?

1. If possible, always randomly assign interventions to experimental units. Randomization ensures, on average, that there is balance for unknown confounders between interventions
2. A confounder is a variable that is associated with both a response and explanatory variable, and consequently causes a spurious association between them. For example, if all samples for intervention A were stored in incubator 1 and all samples for B were stored in incubator 2, and the incubators were found to be operating at different temperatures, then are the observed effects on the outcome due to the interventions or the differences in temperature between incubators? We do not know, as the effects of the interventions and temperature (incubators) are fully confounded

3. If there are known confounding factors, it is always a good idea to modify the design to take account of these; e.g. by blocking

4. Blocking involves dividing experimental units into homogenous subgroups (at the start of the experiment) and allocating (randomizing) interventions to experimental units within blocks so that the numbers are balanced; e.g. interventions A and B are split equally between incubators.

5. Blocking a design to protect against any suspected (or unsuspected) effects on the outcomes caused by processing, storage or assessment procedures is always a good idea; e.g. if more than one individual performs assays, or more than one instrument is used then split interventions so as to obtain balance.

6. In general, it is always better to increase the number of sample experimental units than the number of sub–samples. Study power is directly driven by the number of experimental units $n$.

7. Increasing the number of sub-samples $m$ helps to improve the precision of estimation of the sample effect and allows assay error to be assessed, but has only an indirect effect on study power. Usually there is little benefit to be gained by making $m$ much greater than five.

8. If there are two interventions, then it is always best to divide experimental units equally between interventions. If the aim of an experiment is to compare multiple interventions to a standard or control intervention then it is to better to allocate more experimental units to the standard arm of the study. For example, if a third standard arm (S) were added to the study, in addition to A and B, then it would be better (optimal) to allocate samples in the ratio 2:1:1 to interventions S:A:B.

9. All others things being equal, a better design is obtained if the variances of the explanatory variables are increased, as this is likely to provide a larger effect on the study outcomes. For example, suppose A and B were doses of a drug and a higher dose of the drug resulted in a larger value of the primary study outcome. If the doses for A and B were set at the extremes of the normal range, then the effect on the primary outcome is likely to be much larger than if the doses were only marginally different.

10. If a number of design factors are used then try and make sure that they are independent (uncorrelated). For example, the current design has a single design factor comprising two doses of a drug (A and B). If a second design factor were added, e.g. intravenous (C) or oral delivery (D), then crossing the factors such that the experimental samples are split (evenly) between the four combination A.C, A.D, B.C and B.D provides the optimal design. The factors are independent; using the terminology of experimental design, they are *orthogonal*.

# Appendix 3

DOI: https://doi.org/10.7554/eLife.32486.011

## R code for examples

R is an open source statistical software package and programming language (**R Core Team, 2016**; **Ihaka and Gentleman, 1996**) that is used extensively by statisticians across all areas of scientific research and beyond. The core capabilities of R can be further extended by user developed code packages for very specific methods or specialized tasks; many thousands of such packages exist and can be easily installed by the user from The Comprehensive R Archive Network (CRAN) (**CRAN, 2017**) during an R session. Many excellent introductions to the basics of R are available online and from CRAN (**Venables et al., 2017**), so here the focus is on usage for fitting the models described in the main text with notes on syntax and coding restricted to implementation of these only. A script is available at **Parsons, 2017** to replicate all the analyses reproduced here.

The first dataset considered here is that for the *adjuvant radiotherapy and lymph node size in colorectal cancer* example. For small studies such as this, data can be entered manually into an R script file, by assigning individual observed data variables to a number of named vectors, using the <- operator, and combining together into a data frame (data.frame function), which is the simplest R object for storing a series of data fields which are associated together.

```
> LNsize < c(1.71, 1.72, 1.98, 1.98, 1.88, 1.85, 2.51, 2.98,
   2.55,  3.20,  2.65,  2.80,  1.69,  1.82,  1.72,  1.97,
   1.80,  1.73,  1.72,  2.50,  1.78,  2.65,  2.04,  2.77,
   3.32,  3.11,  3.27,  3.03,  3.07,  3.11,  2.33,  2.86,
   2.48,  2.87,  2.53,  2.52,  2.37,  2.36,  2.36,  2.62,
   2.20,  2.60,  1.33,  1.90,  1.35,  1.87,  1.15,  1.85,
   1.70,  2.07,  1.78,  1.76,  1.78,  1.85,  2.23,  2.50,
   2.14,  2.33,  2.21,  2.16,  2.10,  2.11,  1.89,  2.16,
   1.75,  2.12,  2.58,  2.77,  2.54,  2.65,  2.59,  2.60)
> Subject <- factor(rep(1:12, each = 6), levels = 1:12)
> Sample <- factor(rep(1:2, times = 36), levels = 1:2)
> Slice <- factor(rep(rep(1:3, each = 2), times = 12), levels = 1:3)
> RadioTherapy <- factor(rep(1:2, each = 36), levels = 1:2,
             labels = c("None", "RTShort"))
> LymphNode < data.frame(Subject, Sample, Slice,
             RadioTherapy, LNsize)
```

The *factors* define the design of the experiment, and are built using the rep function that allows structures to be replicated in a concise manner. The first 6 rows of the data frame LymphNode can be examined using the head function.

```
> head(LymphNode, n = 6)
   Subject  Sample  Slice  RadioTherapy  LNsize
   1        1       1      1              None      1.71
   2        1       2      1              None      1.72
   3        1       1      2              None      1.98
   4        1       2      2              None      1.98
   5        1       1      3              None      1.88
   6        1       2      3              None      1.85
```

This is the standard rectanguler form that will be familiar to those who use other statistical software packages or spreadsheets for data storage. More generally data can be read (imported) into R from a wide range of data formats; for instance if data were laid out as above in a spreadsheet programme it could be saved in comma separated format (csv) (e.g. data.csv) and read into R using the following code LymphNode <- read.csv("data.csv"). Naive analysis of

data LymphNode would be implemened using the t.test function

```
> t.test(LNsize ~ RadioTherapy, var.equal = TRUE, data = LymphNode)
    Two Sample t-test
data: LNsize by RadioTherapy
t = 2.501, df = 70, p-value = 0.01473
alternative hypothesis: true difference in means is not equal to 0
95 percent confidence interval:
0.05721632 0.50778368
sample estimates:
mean in group None mean in group RTShort
    2.402778     2.120278
```

This is equivalent to fitting a linear regression model using the R linear model function lm, other than a change in the direction of the differencing of the group means. The R formula notation y ~ x symbolically expresses the model specification linking the response variable y to explanaory variable x; here the response variable is lymph node size LNsize and the explanatory variable is the radiotheraphy treatment RadioTherapy. A full report of the fitted model object mod can be seen using the summary(mod) function. For brevity, the full output is not shown here, but rather individual functions are used to display particular aspects of the fit; e.g. for coefficients coef(mod), confidence intervals confint(mod) and an analysis of variance table anova(mod).

```
> mod <- lm(LNsize ~ RadioTherapy, data = LymphNode)
> anova(mod)
Analysis of Variance Table
Response: LNsize
          Df   Sum Sq   Mean Sq  F value  Pr(>F)
RT.means   1   0.23942  0.23942  1.0868   0.3217
Residuals 10   2.20293  0.22029
---------
Signif. codes:  0  ***  0.001  **  0.01  *  0.05  .  0.1    1
> cbind(coef(mod), confint(mod))
                                2.5%       97.5%
(Intercept)         2.402778  2.2434782  2.56207740
RadioTherapyRTShort -0.282500 -0.5077837 -0.05721632
```

Th analysis by subject proceeds by first calculating lymph node size means for each subject, LNsize.means, using the tapply and mean functions, prior to fitting the linear model, including the new RT.means factor. There is now no need to specify a data frame using the data argument to lm, as response and explanatory variables are newly created objects themselves, so R can find them without having to look within a data frame, as was the case for the previous model.

```
> LNsize.means <- tapply(LymphNode$LNsize, list(LymphNode$Subject),
            mean, na.rm = TRUE)
> RT.means <- factor(rep(1:2, each = 6), levels = 1:2,
            labels = c("None", "RTShort"))
> mod.lm <- lm(LNsize.means ~ RT.means)
> anova(mod.lm)
Analysis of Variance Table
Response: LNsize.means
          Df   Sum Sq   Mean Sq  F value  Pr(>F)
RT.means   1   0.23942  0.23942  1.0868   0.3217
Residuals 10   2.20293  0.22029
> cbind(coef(mod.lm), confint(mod.lm))
```

```
                               2.5%       97.5%
(Intercept)          2.402778  1.975837  2.829718
RT.meansRTShort     -0.282500 -0.886285  0.321285
```

The linear mixed-effects package nlme must be installed before proceeding to model fitting. The model syntax for fitting these models is similar to standard linear models in most respects, with the addition of a `random` argument to describe the structure of the data. Full details of how to specify the model can be found in standard texts such as (***Pinheiro and Bates, 2000***). Confidence intervals of fixed and random effects are provided using the intervals command.

```
> install.packages("nlme")
> library(nlme)
> mod.lme <- lme(LNsize ~ RadioTherapy,
        random = ~1 | Subject / Sample, data = LymphNode)
> anova(mod.lme)
            numDf  denDF   F value    p-value
(Intercept)     1     48   278.60135  <0.0001
RadioTherapy    1     10   1.08682    0.32177
> intervals(mod.lme, which = "fixed")
Approximate 95% confidence intervals
Fixed effects:
                         lower       est.       upper
(Intercept)          2.0175137   2.402778   2.7880418
RadioTherapyRTShort -0.8862853  -0.282500   0.3212853
attr(,"label")
[1]"Fixed effects:"
> intervals(mod.lme, which = "var-cov")
Approximate 95% confidence intervals
Random Effects:
Level:  Subject
                      lower       est.      upper
sd((Intercept))   0.2619509   0.4364928  0.7273346
Level: Sample
                      lower       est.      upper
sd((Intercept))   0.1509095   0.2335944  0.3615832
Within-group standard error:
                      lower       est.       upper
sd((Intercept))   0.09995826  0.12209407  0.14913186
```

Competing models can be compared using likelihood ratio tests.

```
> mod0.lme <- update(mod.lme, random = ~1 | Subject)
> anova(mod0.lme, mod.lme)
          Model  df   AIC     BIC      logLik   Test    L.Ratio  p-value
mod0.lme      1   4   108.71  117.71   -50.36
mod.lme       2   5   -3.9473  7.2952  6.97    1 vs 2   114.66   <0.0001
```

Model fit can be explored using a range of diagnostic plots. For instance, standardized residuals versus fitted values by subject,

```
> plot(mod.lme, resid(., type = "response") ~ fitted(.)
                   | Subject, abline = 0))
```

observed versus fitted values by subject,

```
> plot(mod.lme, LNsize ~ fitted(.)  | Subject, abline = c(0,1))
```

box-plots of residuals by subject,

```
> plot(mod.lme, Subject resid(.), aspect = 1)
```

and quantile-quantile plots.

```
> qqnorm(resid(mod.lme, type = "response"), pch = 19, col = 1,
          main = NULL, las = 1)
> qqline(resid(mod.lme, type = "response"), lty = 2)
```

For the sake of exposition, creating an unbalanced dataset from the original LymphNode data is achieved by randomly removing some data values and re-fitting the mixed-effects model.

```
> set.seed(8845391)
> remove.cells <- sample(1:72, 36, replace=FALSE)
> Unbalanced.LymphNode <- LymphNode[setdiff(1:72, remove.cells),]
> umod.lme <- lme(LNsize ~ RadioTherapy,
    random = ~1 | Subject / Sample, data = Unbalanced.LymphNode)
> anova(umod.lme)["RadioTherapy",]
              numDf   denDF     F value   p-value
RadioTherapy    1         9    0.8122484   0.3909
> intervals(umod.lme, which = "fixed")[["fixed"]][2,]
    lower         est.          upper
-0.9217043   -0.2625918     0.3965206
> intervals(umod.lme, which = "var-cov")
Approximate 95% confidence intervals
Random Effects:
Level: Subject
                     lower        est.        upper
sd((Intercept))   0.2235331   0.4213178   0.7941049
Level: Sample
                     lower        est.        upper
sd((Intercept))   0.1595688   0.2793512   0.4890497
Within-group standard error:
                     lower        est.        upper
sd((Intercept))   0.08837785   0.12387508   0.17362988
```

A subject-based analysis ignores the differences in precision of estimation of means between subjects.

```
> UB.LNsize.means <- tapply(Unbalanced.LymphNode$LNsize,
    list(Unbalanced.LymphNode$Subject), mean, na.rm = TRUE)
> umod.lm <- lm(UB.LNsize.means ~ RT.means)
> cbind(coef(umod.lm), confint(umod.lm))
                                  2.5%        97.5%
(Intercept)       2.3557222   1.9018505   2.8095940
RT.meansRTShort  -0.1990222  -0.8722228   0.4741784
```

The second dataset considered here is grouped binary data from the *lymph node count* example; NA indicates a missing value. For model fitting the non-missing data can be found using the subset and complete.cases functions.

```
> LN.ind <- c(4,  3,  2,  2,  3,  2,  1,  1,  1,  2,  4,  3,
        4,  4,  3,  4,  4,  5,  0,  2,  0,  1,  2,  4,
        NA,  5,  3,  1,  4,  5,  0,  NA,  1,  4,  4,  3,
        NA,  2,  2,  2,  3,  3,  0,  NA,  0,  0,  3,  NA,
```

```
          NA,  NA,  NA,  1,  5,  3,  0,  NA,  2,  2,  3,  NA)
> Subject <- factor(rep(1:12, times = 5), levels = 1:12)
> Sample <- factor(rep(1:5, each = 12), levels = 1:5)
> nRoI <- rep(5, 60)
> RadioTherapy <- factor(rep(rep(1:2, each = 6),times=5), levels = 1:2,
    labels = c("None", "RTShort"))
> nRoI <- rep(5, 60)
> gLymphNode < data.frame(Subject, Sample, RadioTherapy,
    gLNind = as.numeric(LN.ind) / 5, nRoI)
> gLymphNode <- subset(gLymphNode, complete.cases(gLymphNode))
```

Fitting a conventional logistic regression model to the data provides a naive analysis, with estimated coefficients that are log odds-ratios. The glm command indicates that a *generalized linear model* is fitted, with distributional properties identified using the family argument, which for binary data is canonically the binomial distribution with logit link function.

```
> log.reg <- glm(gLNind ~ RadioTherapy, data = gLymphNodeInd,
   family = binomial("logit"), weight = nRoI)
> anova(log.reg, test = "Chisq")
Analysis of Deviance Table
Model:  binomial, link: logit
Response:  gLNind
Terms added sequentially (first to last)
               Df   Deviance Resid.   Df Resid.   Dev      Pr(>Chi)
NULL                                   49  113.299
RadioTherapy  1         21.046         48  92.253   4.485e-06***
---------
Signif. codes: 0 '***' 0.001 '**' 0.01 '*' 0.05 '.' 0.1' ' 1
> cbind(exp(coef(log.reg)), exp(confint(log.reg)))
Waiting for profiling to be done...
                                   2.5%      97.5%
(Intercept)         1.7173913 1.1997511  2.4872605
RadioTherapyRTShort 0.3053412 0.1805858  0.5096359
```

Fitting linear mixed-effects models for non-normal data requires the lme4 package. Model set-up and syntax for lme4 is similar to nlme; for details of implementation for lme4 see (***Bates et al., 2015***) and the vignettes provided with the package.

```
> install.packages("lme4")
> library(lme4)
> gmod.lme4 <- glmer(gLNind ~ RadioTherapy + (1 | Subject), data = gLymphNo-
deInd, family = binomial("logit"), weight = nRoI)
> mod.sum <- summary(gmod.lme4)
> mod.sum[["coefficients"]]["RadioTherapyRTShort","Pr(>|z|)"]
0.01592349
> par.CI <- confint(gmod.lme4, method = "Wald")
> cbind(exp(fixef(gmod.lme4)), exp(par.CI[2:3,]))
                                   2.5%       97.5%
(Intercept)         1.8822378 0.87036620  4.0704923
RadioTherapyRTShort 0.2569425 0.08511571  0.7756438
```

Predictions for the fitted model can be obtained for new data using the predict function, here with no random effects included.

```
> predict(gmod.lme4, newdata = data.frame(RadioTherapy = c("None",
    "RTShort"), type = "response", re.form = ~0)
```

```
       1        2
0.3259761  0.6530474
```

The standard errors of the radiotherapy effects for the conventional logistic regression and mixed-effects model are obtained from the variance-covariance matrices of the fitted model parameters using the vcov function.

```
> sqrt(vcov(log.reg)["RadioTherapyRTShort", "RadioTherapyRTShort"])
0.2642883
> sqrt(vcov(gmod.lme4)["RadioTherapyRTShort", "RadioTherapyRTShort"])
0.5637047
```

# Appendix 4

DOI: https://doi.org/10.7554/eLife.32486.012

## Mathematical description of the naive analysis

The standard method of analysis for simple designed experiments is analysis of variance (ANOVA), which uses variability about mean values to assess significance, under an assumed approximate Normal distribution. Focussing on samples as experimental units, it is decided to collect $m$ replicate measurements of an outcome $y$ on each of $T \times N$ samples, divided into $T$ equally sized treatment groups. Indexing outcomes as $y_{ijt}$, where $i = 1, \ldots, N$, $j = 1, \ldots, m$ and $t = 1, \ldots, T$, the total sums-of-squares (deviations around the mean) which sumarises overall data variability is

$$SS_{Total} = \sum_i \sum_j \sum_t (y_{ijt} - \bar{y}_{...})^2$$

where the overall (grand) mean is $\bar{y}_{...} = \frac{1}{TNm} \sum_i \sum_j \sum_t y_{ijt}$. The Treatment sums-of-squares (SS) is that part of the variation due to the interventions and is given by

$$SS_{Treat} = mN \sum_t (\bar{y}_{..t} - \bar{y}_{...})^2$$

where the treatment means are given by $\bar{y}_{..t} = \frac{1}{Nm} \sum_i \sum_j y_{ijt}$. The residual or error SS is given by

$$SS_{Error} = \sum_i \sum_j \sum_t y_{ijt}^2 - mN \sum_t \bar{y}_{..t}^2$$

and is such that $SS_{Total} = SS_{Treat} + SS_{Error}$. This error SS can be partitioned into that between samples

$$SS_{Error.Samples} = m \sum_i \sum_t y_{i.t}^2 - mN \sum_t \bar{y}_{..t}^2$$

and that within samples

$$SS_{Error.Within} = \sum_i \sum_j \sum_t y_{ijt}^2 - m \sum_i \sum_t y_{i.t}^2$$

where the sample means are given by $\bar{y}_{i.t} = \frac{1}{m} \sum_j y_{ijt}$ and $SS_{Error} = SS_{Error.Samples} + SS_{Error.Within}$. In a naive analysis, ignoring the sampling structure, significance between treatments is *incorrectly* assessed using an F-test of the ratio of the treatment mean-square $MS_{Treat} = SS_{Treat}/(T-1)$ to the error mean-square $MS_{Error} = SS_{Error}/T(Nm-1)$ on $T-1$ and $T(Nm-1)$ degrees of freedom. However, the *correct* analysis is that which uses an F-test of the ratio of the treatment mean-square $MS_{Treat}$ to the between samples error mean-square $MS_{Error.Samples} = SS_{Error.Samples}/T(N-1)$ on $T-1$ and $T(N-1)$ degrees of freedom.

This analysis uses the variability between samples *only* to assess the significance of the treatment effects. The naive analysis pools variability between and within samples and uses this to assess the treatment effects. The naive analysis is generally the default analysis obtained in the majority of statistics software, such as R, if the error structure is not *specifically* stated in the call to analysis of variance.

