## [Decision Letter]

Thank you for submitting your article "Unit of analysis issues continue to be a cause of concern in reporting of laboratory-based research" for consideration by *eLife*. Your article has been reviewed by three peer reviewers, and the evaluation has been overseen by Mark Jit as the Reviewing Editor and Peter Rodgers as the *eLife* Features Editor. The following individuals involved in review of your submission have agreed to reveal their identity: Jenny Barrett (Reviewer #2); Chris Jones (Reviewer #3).

The reviewers have discussed the reviews with one another and the Reviewing Editor has drafted this decision to help you prepare a revised submission.

The reviewers and editors were in agreement on the value of the concept and approach of the manuscript. There were a large number of issues that we felt needed to be addressed, but we do not believe that any of them will take a long time to complete.

Summary:

The tutorial describes issues related to non-independence in data from laboratory and other experiments and further show how they may be overcome, both in a simple way (using subject-level averages) and a more comprehensive way (using mixed models). This is a common problem, and the paper does a good job of both explaining it and giving researchers the tools to deal with it. Its utility is greatly enhanced by very clear detailed illustrative examples and R code to carry out the analyses discussed.

Title:

The current title indicates that the paper is going to show that "Unit of analysis issues continue to be a cause of concern in reporting of laboratory-based research", but that is not what the paper does. Rather, the paper provides guidelines on how to understand the concept of "Unit of analysis" and analyse experiments appropriately. The title should be changed to reflect this.

Essential revisions:

Currently the article contains no guidance on sample size calculation for either the "simple" analyses or the more complex analyses. Nor does it contain any guidance on minimal sample size for the modelling methods suggested. Some comments on sample size and power would be valuable as these are issues that are often neglected by lab scientists. It would also be useful for anyone considering more complex analyses to have an idea of the minimum sample size that can realistically be used to fit the models.

Subsection “Design”. Different designs. Please include some examples of experiments for each situation, as this would make it easier for lab scientists to recognise their type of sample in this list. The example of groups of subjects seems to refer to situations where interest is in the group itself. A common situation instead is where interest is on the effect of treatment on an individual (the experimental unit), but the individuals happen to be grouped (correlated), and it could be useful to clarify this distinction. For example, in laboratory studies the samples may have been analysed in different batches.

Appendix 2 in its current form may not be very helpful or informative to the majority of readers. It does not really explain how to choose among alternative designs, and the equations are likely to be forbidding to non-statisticians. While there are no space limitations in *eLife*, it should be rewritten to focus on the design issues: when should you get more measurements per subject, vs. more subjects? What good are such within-subject replicates (e.g. small improvements in precision, but particularly to be able to measure assay error). It would also benefit from a box summarising what it is showing in a couple of simple sentences, so people that can't get through the equations can at least understand the point it is making.

The code in Appendix 3 is very helpful, but it is difficult to read in its present form. We recommend publishing it in text form using indentation, colours, and explanatory text interspersed with the sections of code to explain it. Ideally, it should be written as a tutorial (with portions of text and code interspersed).

It would also be good to show how the data for each of the examples is structured within a database – i.e. with variables representing the individual, clustering, groupings etc. Lab scientists are generally less familiar with how data is entered/stored in databases/stats software, and they may be familiar with GraphPad Prism, which accepts data in very different formats to the standard format required for the analyses presented in this paper. Appendix 3 could be expanded to include the data frames next to the R code (at the start of each example).

---

## [Author Response]

Title:The current title indicates that the paper is going to show that "Unit of analysis issues continue to be a cause of concern in reporting of laboratory-based research", but that is not what the paper does. Rather, the paper provides guidelines on how to understand the concept of "Unit of analysis" and analyse experiments appropriately. The title should be changed to reflect this.

Title changed to “Unit of analysis issues in laboratory based research: a review of concepts and guidance on study design and reporting”.

Essential revisions:Currently the article contains no guidance on sample size calculation for either the "simple" analyses or the more complex analyses. Nor does it contain any guidance on minimal sample size for the modelling methods suggested. Some comments on sample size and power would be valuable as these are issues that are often neglected by lab scientists. It would also be useful for anyone considering more complex analyses to have an idea of the minimum sample size that can realistically be used to fit the models.

A new subsection has been added, after the ‘Analysis’ subsection, that discusses sample size estimation from initially a very simple design, to more complex GLMMs via simulation.

Subsection “Design”. Different designs. Please include some examples of experiments for each situation, as this would make it easier for lab scientists to recognise their type of sample in this list. The example of groups of subjects seems to refer to situations where interest is in the group itself. A common situation instead is where interest is on the effect of treatment on an individual (the experimental unit), but the individuals happen to be grouped (correlated), and it could be useful to clarify this distinction. For example, in laboratory studies the samples may have been analysed in different batches.

Simple examples have been added to the design types in the subsection “Design”. The ‘Groups of subjects’ example has been expanded to cover the kind of ‘batch-effects’ identified by the reviewer.

Appendix 2 in its current form may not be very helpful or informative to the majority of readers. It does not really explain how to choose among alternative designs, and the equations are likely to be forbidding to non-statisticians. While there are no space limitations in eLife, it should be rewritten to focus on the design issues: when should you get more measurements per subject, vs. more subjects? What good are such within-subject replicates (e.g. small improvements in precision, but particularly to be able to measure assay error). It would also benefit from a box summarising what it is showing in a couple of simple sentences, so people that can't get through the equations can at least understand the point it is making.

Appendix 2 has been modified to discuss fundamental design issues for a putative example experiment. It now focuses more on design issues, and uses less mathematical language that should be more accessible to readers of *eLife*. The mathematical details of the (incorrect) naïve analysis has been moved to a separate new appendix (Appendix 4).

The code in Appendix 3 is very helpful, but it is difficult to read in its present form. We recommend publishing it in text form using indentation, colours, and explanatory text interspersed with the sections of code to explain it. Ideally, it should be written as a tutorial (with portions of text and code interspersed).

Appendix 3 (R code for examples) has been completely revised and re-written along the lines suggested here. It is now written in the style of a tutorial with code indented and coloured to distinguish it from the main text. R output is also now provided to help those wishing to check exactly what would be produced if the code were pasted directly into R.

It would also be good to show how the data for each of the examples is structured within a database – i.e. with variables representing the individual, clustering, groupings etc. Lab scientists are generally less familiar with how data is entered/stored in databases/stats software, and they may be familiar with GraphPad Prism, which accepts data in very different formats to the standard format required for the analyses presented in this paper. Appendix 3 could be expanded to include the data frames next to the R code (at the start of each example).

We agree that the data entry in the previous example R code was not realistic. Appendix 3 now explicitly shows the format of the data in R. A note is also added to explain how data would normally be entered using the read statement that will import data into R from standard spreadsheets or databases.